# OPriv: Optimizing Privacy Protection for Network Traffic

**Louma Chaddad** *, **Ali Chehab**  and **Ayman Kayssi**

Department of Electrical and Computer Engineering, American University of Beirut, Beirut 1107 2020, Lebanon;
chehab@aub.edu.lb (A.C.); ayman@aub.edu.lb (A.K.)
* Correspondence: lac07@mail.aub.edu

**Abstract:** Statistical traffic analysis has absolutely exposed the privacy of supposedly secure network traffic, proving that encryption is not effective anymore. In this work, we present an optimal counter-measure to prevent an adversary from inferring users' online activities, using traffic analysis. First, we formulate analytically a constrained optimization problem to maximize network traffic obfuscation while minimizing overhead costs. Then, we provide OPriv, a practical and efficient algorithm to solve dynamically the non-linear programming (NLP) problem, using Cplex optimization. Our heuristic algorithm selects target applications to mutate to and the corresponding packet length, and subsequently decreases the security risks of statistical traffic analysis attacks. Furthermore, we develop an analytical model to measure the obfuscation system's resilience to traffic analysis attacks. We suggest information theoretic metrics for quantitative privacy measurement, using entropy. The full privacy protection of OPriv is assessed through our new metrics, and then through extensive simulations on real-world data traces. We show that our algorithm achieves strong privacy protection in terms of traffic flow information without impacting the network performance. We are able to reduce the accuracy of a classifier from 91.1% to 1.42% with only 0.17% padding overhead.

**Keywords:** traffic masking; privacy; optimization; information theory; obfuscation; information leakage



## 1. Introduction

Nowadays, statistical traffic analysis is becoming an attractive tool for developing algorithms to evaluate and manage internet network traffic. By definition, traffic analysis is a network engineering technique that consists of examining statistical features of flow packets (e.g., packet sizes, inter-arrival times, and packet directions) and building classifiers, using machine-learning algorithms to infer traffic information. Network administrators take advantage of traffic analysis techniques for the purposes of intrusion detection to detect abnormal traffic [1–3], and to differentiate malicious traffic flows [4]. In addition, a proper deployment of traffic analysis provides valuable insights for resource management, traffic control, diagnostic checking, and provisioning. Engineers can use this information to build robust networks and avoid possible delays. To this end, traffic analysis is used to support internet-based services, including banking, health, military, government, electrical systems, and transportation.

On the other hand, traffic analysis can be misused to launch attacks to infer knowledge on network traffic by means of exploiting side-channel information leakage. Even with the growing use of encryption to protect traffic content, traffic analysis is becoming a common attack, threatening privacy, anonymity, and confidentiality. These types of attacks can cause major privacy breaches on military systems, banking, healthcare, etc.

In fact, an adversary can misuse traffic analysis in order to conclude a user's online activity, which is typically private and includes sensitive information. These privacy breaches, also referred to as 'side-channel information leaks' range from predicting users' locations [5], distinguishing the downloaded web pages [6]; identifying language in an

encrypted VoIP conversation [7]; obtaining records of an encrypted VoIP chat [8]; or identifying critical information about the underlying type of applications [9–11].

In brief, traffic analysis nowadays is considered a major threat of confidentiality of network applications. Additionally, tools for protecting information traffic privacy still fall short of what is required in terms of strong privacy protection and low performance overhead. In this work, we aim to answer the following question: can we optimally protect users' privacy without deteriorating internet traffic behavior?

This work is an extension and improved version of our previous work [12] to fully address the problem of optimizing network obfuscation. We aim for a traffic obfuscation system that strikes a good balance between efficiency in obstructing application tracing and maintaining minimal performance impact. Our goal is to find the sufficient padding or fragmentation that guarantees good secrecy while optimizing the overhead. In addition, we present a structured and comprehensive overview into entropy-based obfuscation systems in order to ensure security and efficiency at the same time. The main contributions of this work are as follows:

- Verification of the benefits achieved by an optimal solution for the mathematical problem of network traffic obfuscation problem;
- Suggestion of information theoretic metrics for quantitative privacy measurement, using entropy;
- Proposal of OPriv, a practical solution based on the IBM ILOG CPLEX optimization package (ILOG);
- Verification of the effectiveness of OPriv via two criteria, using both analytical and experimental models;
- Conducting a comparative study between OPriv and state-of-the-art obfuscation models.

The rest of this paper is organized as follows. We summarize the related work in Section 2. Section 3 presents the threat model, the design of the obfuscation system, and the experiments to evaluate the efficiency of the system. In Section 4, we present new metrics based on entropy to enhance obfuscation systems. In Section 5, we suggest a measure to evaluate our proposed obfuscation model and compare it with other techniques. Section 6 discusses the implication issues. Finally, we conclude the paper in Section 7.

## 2. Literature

The study of traffic analysis for network data flow is not new. Given the importance of traffic classification for purposes of security and quality of service (QoS) management, different machine learning solutions have been used effectively [2,13–17].

Conversely, there is currently a new research direction focusing on traffic obfuscation to thwart classification in order to protect users' security. Different systems have been developed to ensure confidentiality, performance, and efficiency of networks. In the following, we group the classification-obfuscation methods into four types: anonymization, mutation, morphing, and tunneling. We classify some of the reviewed work in Table 1.

**Table 1.** Traffic obfuscation techniques.

| Work | Year | Attacker Model | Acc. Before | Defense Method | Acc. After | Defense Approach |
|------|------|----------------|-------------|----------------|------------|-------------------|
| [18] | 2003 | Bayes | 100% | Dummy packets insertion | 50% | Anonymization |
| [19] | 2012 | Liberatore and Levine | 87% | MTU padding | 41% | Mutation |
| [19] | 2012 | Herrmann et al. | 98% | Random padding | 40% | Mutation |
| [19] | 2012 | Herrmann et al. | 98% | Random MTU padding | 11% | Mutation |
| [19] | 2012 | Herrmann et al. | 98% | Linear padding | 73% | Mutation |
| [19] | 2012 | Herrmann et al. | 98% | Exponential padding | 61% | Mutation |
| [20] | 2012 | Naive Bayes | 65% | Elephants and mice padding | 28% | Mutation |
| [21] | 2009 | Naive Bayes | 98% | Convex Optimization | 63% | Morphing |
| [22] | 2013 | SVM and NN | 83% | Traffic demultiplexing | 44% | Other |
| [23] | 2016 | K-NN | 91% | Adaptive padding | 20% | Mutation |
| [24] | 2018 | CNN model | 98% | Random padding | 60% | Mutation |
| [25] | 2018 | Optimal attacker | 94% | DynaFlow | 44% | Morphing and Mutation |
| [26] | 2018 | Deep learning | 90% | BuFLO | 12% | Mutation |
| [27] | 2006 | Naive Bayes | 98% | MTU padding | 7% | Mutation |
| [28] | 2011 | SVM | 80% | Camouflage | 4% | Anonymization |

Anonymization consists of hiding key information that provides important information for traffic classification, such as IP addresses, port numbers, and MAC addresses. In this context, multi-path routing [29,30] and NATing [31] can be used for anonymizing the communicated traffic. TOR is a well-known system for anonymous communication based on the second generation of the onion routing model [32]. However, in the recent past, it was verified that even such systems are vulnerable to detection by machine-learning techniques [33–35] which have proved efficient in classifying TOR traffic. Other proposed anonymization techniques consist of inserting additional dummy packets and transmitting them to conceal the real traffic. For instance, ref. [18] proposed the use of heavy traffic to hinder the adversary's ability to tamper with the links. Additionally, the proposed defenses in ref. [36–38] deliberately drop some packets, known as defensive dropping, and in ref. [39], inject artificial delay intentionally.

Traffic mutation relies on changing the flows' statistical characteristics to confuse a classifier and to make it difficult to identify the original traffic [40]. It consists of properly modifying the packet sizes and/or the packet interarrival times (IAT), using padding, fragmentation, and buffering; consequently, the statistics of the overall conveyed traffic become considerably dissimilar from the original one. In this context, padding and fragmentation are used to hide packet size information. On the other hand, buffering aims to hide the interarrival time information. These methods have a great impact in reducing the accuracy of statistically-based traffic classifiers. Maximum transmission unit (MTU) padding, for example, is a well-known mutation technique that consists of padding all the flow packets to the maximum payload size MTU [19]. Random padding is another technique that consists of randomly padding the packets [41].

However, mutation and anonymization methods can result in a drastic overhead in terms of the amount of data being sent [42]. This leads to performance degradation in terms of delay and bandwidth consumption [43]. On the other hand, some works proved that it is still possible to classify the traffic, even when implementing these methods [44].

Morphing techniques aim at confusing the classifier into classifying the target application traffic as another type. Wright et al. proposed a morphing technique that consists of transforming one class of traffic to look like another class by applying convex optimization [20,21]. Nevertheless, this method applies to devices with strong computational capabilities. In addition, it increases latency, due to the generation of random numbers for each input packet.

Tunneling hides packet-related features by the use of encryption and the creation of virtual networks. The use of VPN, for example, ensures hiding the connection metadata and, consequently, the users' privacy. However, the work in [45] categorizes VPN traffic based on the application name and traffic type. Ref. [22] suggests using virtualized MAC addresses to obscure the adversary's analysis. Their model consists of creating multiple virtual MAC interfaces over a single wireless card. Then, the packets over these interfaces are dynamically scheduled, and the packet features are reshaped over each virtual interface. However, this solution is more appropriate for using a Wi-Fi connection on a computer.

We proposed in our previous works, so far, a handful of obfuscation algorithms resulting in significant outcomes [46–49]. Our goal was to address the privacy protection of network communication against side information based inferential attacks by exploring a combination of concepts, specifically anonymization, mutation, and morphing.

In ref. [12], we presented a mathematical model for network obfuscation. Then, we formulated analytically the problem of selecting both the target app from a data set and the packet length from the designated target app to mutate to.

Despite the efficient results of our earlier obfuscation algorithms, we aim in this work to analyze mathematically our previous results to find a systematic explanation for them. This can be realized using concepts of information theory. Information entropy is a mathematical concept that measures how much information there is in an event [50]. It was widely implemented in security systems to detect traffic anomalies [51,52], or for systems of DDoS defense [53]. To the best of our knowledge, the concept of measure function

that helps in explaining and predicting theoretically the obfuscation results for a specific defense system has not been tackled yet in the literature. In this work, we aim to provide a better understanding of the implementation of entropy-based methods in obfuscation systems against traffic analysis attacks.

### 3. Traffic Privacy Protection

*3.1. Adversary Model*

We consider a passive adversary who uses machine-learning methods to trace end-user application identification through network traffic classification. In wireless LANs, eavesdropping is easy, due to the shared medium; hence, encrypted traffic samples that users send over wireless links are practically exposed to sniffers monitoring the traffic. Even worse, the attacker can have direct access to the encrypted data of a user from the server. This adversary could be, for example, a government that gained access to an ISP. Hence, the adversary has flow features information, such as timing, size, direction, and count of packets in a specific encrypted network flow.

We illustrate in the reference scenario model in Figure 1, an adversary who monitors the internet connection of a mobile user. The adversary uses sniffer software (e.g., Wireshark) and does not have any knowledge about the software or encryption schemes that the user implements. The classification system collects traffic traces of the user and identifies which mobile network application they are running. We use this traffic classification attack scenario to test our obfuscation system model in Section 3.5.

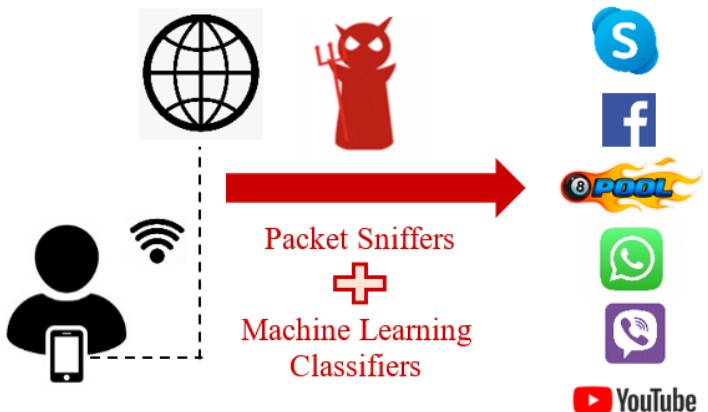

**Figure 1.** Mobile traffic classification attack.

*3.2. Classification Attack*

We consider six popular network applications, including chatting, online gaming, video calling and streaming, VoIP, and social media browsing. We list the applications and their differences in Table 2, and we label them from 1 to 6. We build four machine learning algorithms as explained in our previous work [46]. The accuracy of the different classifiers and their corresponding parameters are presented in Table 3.

**Table 2.** Experiment data set [49].

| # | App | Type |
|---|-----|------|
| 1 | Skype | Video call |
| 2 | Facebook | Interactive browsing |
| 3 | 8 ball pool | Game |
| 4 | WhatsApp | Messaging |
| 5 | Viber | VOIP |
| 6 | YouTube | Video streaming |

**Table 3.** Data set accuracy [49].

| Model | Parameters | Accuracy |
|---|---|---|
| SVM | Quadratic kernel | 76.7% |
| Bagged trees | Minimum leaf size = 2 | 90.0% |
| KNN | Number of neighbors = 5; Distance weight = squared inverse | 83.9% |
| Random forest | Minimum leaf size = 2 | 91.1% |

### 3.3. Obfuscation Optimization Problem

Information leakage of the flow content is provided typically from traffic features, such as packet lengths, inter-packet times, traffic direction, etc.

To achieve maximum protection, we formulated mathematically in [12] the optimization problem with the goal of maximizing obfuscation, while minimizing the overhead and being subject to traffic constraints. We divide the packet length traces in each app traffic set into bins of size $w$. Then, we study the probability of each bin of packet lengths across all instances. We modify each packet in the flows, created by the source app to defend, such that their distribution becomes similar to packet sizes of a different target app.

Consider an application $i$ defined by $[m_i, L_i, P_i]$ where the following is true:

- $m_i$ is the number of possible packet length ranges of size $w$.
- $L_i = [L_i^1, L_i^2, \ldots, L_i^{m_i}]$ is the vector of smallest lengths in each bin of size $w$. The average of $L_i$ is denoted by $\text{Avg}(L_i)$.
- $P_i = [P_i^1, P_i^2, .., P_i^{m_i}]$ is the vector of length probabilities for bins of size $w$.

An incoming packet from the source application can be defined by its size $l$ and the probability $p$ of the bin to which $l$ belongs. We define a binary variable for each application that indicates whether the app is selected as target app or not; $x_i$ denotes the variable for the $i$th app. Given a set of $N$ applications, the vector $X$ (of length $N$) represents $[x_1, x_2, ..., x_N]$.

In addition, we delineate a binary variable for each bin of an application $i$ that indicates whether the packet bin is selected to be mutated to or not. It is worth noting that the smallest size in a bin is used for mutation. $y_i^j$ denotes the variable for the $j$th bin of application $i$. Given a set of $m_i$ ranges, the vector $y_i$ (of length $m_i$) designates $[y_i^1, y_i^2, ..., y_i^{m_i}]$.

In order to minimize the accuracy of the classifier, we need to find the probability of a packet length in a bin of a target app such that it is similar to $p$. At the same time, to minimize the overhead, the selected packet length must be similar to $l$. In the case that the difference between $p$ and the designated probability is too small (less than 0.1 in our case), the difference between the selected length and $l$ needs to be checked that it is less than the average difference. This guarantees that the overhead does not reach a large value.

The obfuscation optimization problem for mutating an incoming packet from a source app is as follows: Find the vector $X$ (of length $N$), and the vector $y_i$ (of length $m_i$), and minimize:

$$Min \sum_{i=1}^{N} x_i \left( \sqrt{\sum_{j=1}^{m_i} y_i^j (\mid P_i^j - p \mid^2 \times \mid L_i^j - l \mid^2)} \right)$$

Subject to the following:

1.

$$P_i^j \in [0,1]; \ i = 1, ..., N$$

2.

$$\mid L_i^j - l \mid < \mid Avg(L_i) - l \mid \ for \mid P_i^j - p \mid < 0.1$$

3.

$$\sum_{i=1}^{N} x_i = 1; \ x_i = 0 \ or \ 1; i = 1, ..., N$$

4.

$$\forall i \in [1, N], \sum_{j=1}^{m_i} y_i^j = 1; y_i^j = 0 \text{ or } 1; j = 1, ..., m_i$$

### 3.4. OPriv

In this part, we present OPriv, a practical solution to the optimization problem. At a high-level, given a source application, our goal is to mutate a source application traffic such that it is confused as some different target application. The user chooses first the source application that they would like to defend, as well as a set of target applications that they would like to make the source application resemble. Of course, we need to make sure that the implemented mutation results in a minimum added number of overhead bytes. OPriv dictates how each packet size from the source application should be mutated such that the resulting distribution resembles the target with a minimum of overhead.

The model has constraints, including a quadratic term and therefore, it is a mixed-integer quadratically constrained program (MIQCP) problem. As a practical result, we deploy and solve the problem deciding on mixed-integer quadratic problem MIQP linearization, using the software IBM ILOG CPLEX Optimization Studio Version: 12.9.0.0. CPLEX is chosen for its robustness in solving large-scale, real-world problems. The coding in CPLEX specifies the problem parameters, optimization variables, objective function, and constraints discussed previously . The result for one iteration is as follows: vector X to identify the target app for mutation, and the corresponding Y to specify the packet bin to mutate to in the target app.

The proposed method, implemented using IBM's CPLEX optimization package (ILOG), requires 1–4 min to solve with a dual-core Intel i7 processor.

To solve several instances of the same problem, we define our model in Java and call Cplex Java API on Eclipse to solve it. We record in Table 4 the time needed to execute each instance of optimization for the different applications.

**Table 4.** Average execution time for each instance of optimization for the different applications.

| Application | Average Execution Time (s) |
|:-----------:|:--------------------------:|
| Skype | 0.16 |
| Facebook | 0.36 |
| Game | 0.2 |
| WhatsApp | 0.14 |
| Viber | 0.36 |
| YouTube | 0.19 |

### 3.5. Obfuscation Experiments and Results

For each incoming packet from the source app, we implement the solution of the optimization problem to select the optimal target app and packet size to mutate to. First, we select the unique values of lengths and their corresponding probabilities of a certain app in our data set as input to our code in IBM ILOG CPLEX. The output results are the optimal packet sizes to mutate to. We generate tables mapping each length source packet to its corresponding optimal target size offline, and so the entire process becomes more practical, inducing minimal latency, especially for dynamic applications. Those mapping tables are the results of our optimization problem and define how we should mutate packets. Skype, Facebook and Game have around 1400 unique values of sizes. WhatsApp and Viber have around 200 unique values. As for YouTube, the number is 1000. For the sake of simplicity, we show only a sample of seven mapping values in Tables 5–10. According to those tables, the difference between the original length and the optimal length to mutate to is minimal, which achieves our goal in minimizing the padding overhead. For each incoming packet, OPriv refers to the appropriate mapping table, matches its packet length and probability using our offline tables and infers the corresponding optimum packet size that we need to

mutate to. Finally, we apply our masking by means of padding and fragmenting. If the incoming packet length is less than the corresponding optimum length , we proceed by padding it with zeroes such that the resultant packet length become similar. Otherwise, we chop the packet into two smaller fragments where one of them has the same size as the corresponding optimum length.

To evaluate OPriv, we use our four model classifiers described in Section 3.2, and we predict the classes of the mutated traffic. We present, in Table 11, comprehensive simulation results of the mutation using OPriv of each application in our data set. For example, mutating Game using OPriv can decrease the SVM classifier's accuracy from 76.7% to 0.85%, Bagged Trees classifier's accuracy from 90% to 1.45%, KNN classifier's accuracy from 83.9% to 1.83%, and Random Forest classifier's accuracy from 91.1% to 1.42% with only 0.17% average overhead. As expected, using optimization, OPriv outperforms all of the other state-of-the-art defense solutions against traffic analysis. It realizes a good balance between efficiency and overhead, especially for Game, WhatsApp, Viber, and YouTube. We explain in what follows the inefficiency of Skype and Facebook mutation.

**Table 5.** Sample of length mapping of Skype App, using OPriv.

| Original Length | Original Length Problem | Optimum Length to Mutate to |
|---|---|---|
| 85 | 0.0086 | 54 |
| 103 | 0.0072 | 154 |
| 120 | 0.008 | 154 |
| 134 | 0.00623 | 154 |
| 145 | 0.004064 | 104 |
| 760 | 0.00031 | 804 |
| 1440 | 0.000431 | 1354 |

**Table 6.** Sample of length mapping of Facebook app using OPriv.

| Original Length | Original Length Problem | Optimum Length to Mutate to |
|---|---|---|
| 66 | 0.33539 | 54 |
| 74 | 0.00487 | 104 |
| 122 | 0.00015 | 154 |
| 187 | 0.00030 | 205 |
| 384 | 0.000335 | 320 |
| 898 | 0.00014 | 854 |
| 1514 | 0.27281 | 1504 |

**Table 7.** Sample of length mapping of Game App, using OPriv.

| Original Length | Original Length Problem | Optimum Length to Mutate to |
|---|---|---|
| 74 | 0.01562 | 104 |
| 78 | 0.01908 | 154 |
| 102 | 0.00757 | 355 |
| 544 | 0.00286 | 704 |
| 363 | 0.00113 | 554 |
| 732 | 0.00105 | 804 |
| 1210 | 0.00186 | 1307 |

**Table 8.** Sample of length mapping of WhatsApp app, using OPriv.

| Original Length | Original Length Problem | Optimum Length to Mutate to |
|:---:|:---:|:---:|
| 66 | 0.43874 | 104 |
| 74 | 0.01267 | 254 |
| 192 | 0.00694 | 154 |
| 292 | 0.00815 | 304 |
| 308 | 0.00724 | 355 |
| 914 | 0.00663 | 1057 |
| 1514 | 0.26705 | 1354 |

**Table 9.** Sample of length mapping of Viber app, using OPriv.

| Original Length | Original Length Problem | Optimum Length to Mutate to |
|:---:|:---:|:---:|
| 50 | 0.0628 | 154 |
| 63 | 0.04657 | 104 |
| 65 | 0.04173 | 304 |
| 150 | 0.02118 | 205 |
| 155 | 0.02219 | 254 |
| 160 | 0.02028 | 154 |
| 592 | 0.00119 | 654 |

**Table 10.** Sample of length mapping of YouTube app, using OPriv.

| Original Length | Original Length Problem | Optimum Length to Mutate to |
|:---:|:---:|:---:|
| 66 | 0.33381 | 62 |
| 78 | 0.01202 | 104 |
| 500 | 0.00562 | 592 |
| 514 | 0.00668 | 240 |
| 548 | 0.00616 | 300 |
| 1484 | 0.01667 | 804 |
| 1514 | 0.43893 | 1354 |

**Table 11.** Results of apps mutation, using OPriv.

| Algorithm | SVM | BT | KNN | RF | Overhead |
|:---:|:---:|:---:|:---:|:---:|:---:|
| Original | 76.7% | 90% | 83.9% | 91.1% | - |
| Skype | 40.90% | 51.21% | 43.68% | 40.92% | 3.45 % |
| Facebook | 42.2% | 47.77% | 47.54% | 41.12% | 4.41% |
| Game | 0.85% | 1.45% | 1.83% | 1.42% | 0.17% |
| WhatsApp | 11.25% | 12.93% | 10.8 % | 10.83% | 0.93% |
| Viber | 13.2% | 12.58% | 11.74% | 11.78% | 0.84% |
| YouTube | 8.97% | 2.58% | 8.49% | 9.1% | 1.58% |

## 4. Entropy Concepts to Enhance Obfuscation Efficiency

Shannon entropy and Kullback–Leibler divergence (also known as information divergence or relative entropy) are perhaps the two most important quantities in information theory and its applications. Most of these measures have never found any application in machine learning and network obfuscation systems. In an attempt to explain the efficiency of our obfuscation systems, we present in what follows the metrics of information to test their performance, namely, degree of traffic masking effectiveness, Kullback–Leibler divergence, and traffic divergence.

*4.1. Degree of Traffic Masking Effectiveness*

Given a random variable $X$, which takes values of the finite set of $M$ values $x_1, x_2, \ldots, x_M$ and $p_i := P(X = x_i)$ is the probability of occurrence of $x_i$; hence, the Shannon entropy is as follows:

$$H = -\sum_{i=1}^{M} p_i log(p_i)$$

The conditional entropy of a variable $Y$ given $X$ of $M$ values is defined by the following:

$$H(Y/X) = \sum_{i=1}^{M} p(X = i)H(Y/X = i)$$

Consider a traffic network and its set of $p$ features $X = X_1, X_2, \ldots, X_p$, the degree of traffic masking effectiveness using a certain obfuscation algorithm that mutates any of the $p$ features, can be studied in terms of conditional entropy.

Most traffic analysis attacks use information derived from the sequences of packet lengths, and/or from interarrival times as input. This motivates our definition of degree of traffic masking effectiveness to measure how much the adversary can learn about a certain network application, given the observed packet lengths or gap times leaked by the traffic.

Let $\Phi$ be the set of packets that refer to a specific application, and let $\Psi$ be the side information on $\Phi$ gained by the adversary by observing the encrypted traffic. The degree of masking effectiveness is measured by the conditional entropy $H(\Phi/\Psi)$ to denote how much information an adversary can learn about the identification of a certain application, given the observed traffic features. Accordingly, the smaller that $H(\Phi/\Psi)$ is, the more likely that $\Phi$ is a function of $\Psi$. This means that mutating or modifying any pattern of $\Psi$ sincerely affects the identification of $\Phi$. Moreover, a high $H(\Phi/\Psi)$ indicates that $\Phi$ is less dependent on $\Psi$. Hence, no matter how much the obfuscation algorithm mutates $\Psi$, that does not significantly affect the fingerprinting of $\Phi$.

As a proof of concept, we study the degree of the masking effectiveness of $H(\Phi/\Psi)$ for the apps in our data set. We evaluate $H(\Phi/\Psi)$ for variables of side information being packet lengths, and packet interarrival time IAT. We present the results in Table 12.

**Table 12.** Degree of traffic masking effectiveness.

| $\Phi$    $\Psi$ | Lengths | IAT |
|---|---|---|
| Skype | 0.3776 | 0.9336 |
| Facebook | 0.4374 | 0.7545 |
| Game | 0.1168 | 0.7530 |
| WhatsApp | 0.0451 | 0.3511 |
| Viber | 0.1862 | 0.6020 |
| YouTube | 0.2635 | 0.4362 |

According to Table 12, the values of $H(\Phi/\Psi)$ with $\Psi$ = packet lengths, are relatively low, indicating a high dependency of our apps on their packet lengths. In this case, any security algorithm that mutates the lengths effectively will obfuscate the adversary knowledge of the exact application, using traffic analysis. The impact of mutation of lengths of Game, WhatsApp, Viber, and YouTube are foreseen and tangible, as $H(\Phi/\Psi)$ for those apps are relatively minor in comparison to those of Skype and Facebook. This can explain the results in Table 11, where OPriv appears to be less efficient in the cases of Skype and Facebook, in comparison to the other apps.

The values of $H(\Phi/\Psi)$ with $\Psi$ = packet IAT are relatively high. This means that in our case of traffic traces, the knowledge of the exact type of application does not mainly depend on IATs. Any obfuscation algorithm that modifies the IAT would be less effective in obscuring the application identity.

In our previous work [47], we proposed *AdaptiveMutate*, a privacy thwarting technique with 3 variations where we mutate the packet lengths, and/ or interarrival times of the source app to defend such that the lengths or IAT of the output packets appear as though they are coming from the target app probability mass function . The simulation results show a significant efficiency of *AdaptiveMutate* when applied using lengths mutation. However, IAT mutation is not efficient enough in reducing the classifier's accuracy. This is proved by the values of $H(\Phi/\Psi)$ in Table 12.

### 4.2. Kullback–Leibler Divergence

In our previous works [49], we presented an obfuscation algorithm that consists of mutating the packet lengths of a source app to those lengths from a target app, having similar bin probability. We used a trial-and-error process where we mutated comprehensively, using our algorithm, each application to all the other apps in order to decide on the best target app to mutate to. We present instead in this part the Kullback–Leibler divergence, a mathematical defined metric to decide on which target app is best to mutate to.

Given two complete discrete probability distributions $P = (p_1, p_2, \ldots, p_n)$, and $Q = (q_1, q_2, \ldots, q_n)$, where $p_i, q_i \in [0,1]$ and $\sum_{i=1}^{n} p_i = \sum_{i=1}^{n} q_i = 1$ . The Kullback–Leibler divergence is a measure of divergence between $P$ and $Q$. According to [54], the Kullback–Leibler $D(P \mid\mid Q)$ is obtained using the following formula:

$$D(P \mid\mid Q) = \sum_{i=1}^{n} (p_i log(\frac{p_i}{q_i}))$$

The Kullback–Leibler divergence can be used as a decision function to decide on which network application in a data set to mutate to. In our context of network obfuscation assessment, we denote by $P$ the probability distribution of a feature in the source app, and by $Q$ that of the destination app to mutate to. The smaller $D(P \mid\mid Q)$ is, the closer the distributions $P$ and $Q$ are, and vice versa. A better obfuscation efficiency would be realized in cases where $Q$ is chosen in such a way that $D(P \mid\mid Q)$ is large. The more that the source and destination app feature probability distributions are different, the more there is changeable features of the source app, and consequently, it would confuse it as the destination app by the classifier. The changeable features make the accurate identification of network application more difficult, whereas, in the case that $P$ and $Q$ are similar, the mutation of $P$ would not change much its distribution, making the confusion less significant.

Table 13 presents KL divergence for lengths probabilities of the different apps and explains the results of our previous works [49] where mutation to target apps with higher values of KL lead to better obfuscation results.

According to Table 14, KL divergence for IAT probabilities estimation of the different apps are close to zero and therefore, are very similar. Hence, there is no changeable features in the case of IAT mutation, making the confusion of a classifier less significant. This is proved in the results of [49] when we try to mutate IATs.

**Table 13.** Kullback–Leibler divergence of packet lengths probabilities.

| P ＼ Q | Skype | Facebook | Game | WhatsApp | Viber | YouTube |
|---|---|---|---|---|---|---|
| Skype | 0 | 1.7775 | 2.5947 | 2.5188 | 2.2243 | 2.6258 |
| Facebook | 1.9620 | 0 | 4.2658 | 3.1426 | 4.2831 | 3.2296 |
| Game | 2.3262 | 4.1728 | 0 | 3.0878 | 2.3356 | 2.1204 |
| WhatsApp | 2.9386 | 3.1240 | 3.1057 | 0 | 3.8656 | 4.2221 |
| Viber | 2.6613 | 2.7377 | 3.6844 | 3.1471 | 0 | 2.3251 |
| YouTube | 2.6019 | 3.1697 | 2.1257 | 4.2374 | 2.8794 | 0 |

**Table 14.** Kullback–Leibler divergence of IAT probabilities.

| P \ Q | Skype | Facebook | Game | WhatsApp | Viber | YouTube |
|---|---|---|---|---|---|---|
| Skype | 0 | 0.0006 | 0.0002 | 0.0226 | 0.0655 | 0.0017 |
| Facebook | 0.0009 | 0 | 0.0018 | 0.0194 | 0.0617 | 0.0006 |
| Game | 0.0002 | 0.0009 | 0 | 0.0231 | 0.0665 | 0.0019 |
| WhatsApp | 0.0727 | 0.0512 | 0.0797 | 0 | 0.0500 | 0.0324 |
| Viber | 0.2863 | 0.2154 | 0.3529 | 0.073 | 0 | 0.1933 |
| YouTube | 0.0033 | 0.0011 | 0.0039 | 0.0158 | 0.0592 | 0 |

*4.3. Traffic Divergence*

Traffic divergence $TD(P \parallel Q)$ between the original and mutated traffic can be expressed in terms of Kullback–Leibler divergence. $P$ denotes in this case the probability distribution of a certain side information before mutation, and $Q$ that after mutation. Hence, $TD(P \parallel Q)$ measures to what extent a certain obfuscation algorithm can modify the probability distribution of a side information of an app traffic. $TD$ reflects the efficiency of the obfuscation algorithm, and on the amount of overhead resulting from it. A high $TD$ indicates a high change of the probability distribution of the specified side information feature. If this feature brings a small degree of traffic masking (detailed in Section 4.1), that results in an efficient protection against traffic analysis. However, a high traffic divergence can be reflected in a higher overhead. $TD$ can be used to compare between obfuscation algorithms. It gives an insight of how much the algorithm causes modification and overhead. The finest obfuscation algorithm is the one that mutates side information with the smallest degree of traffic masking and results in least $TD$ at the same time. Table 15 presents the traffic divergence between the original traffic of our app data set, and the mutated traffic, using OPriv.

**Table 15.** Length traffic divergence $TD$ between original and mutated traffic, using OPriv.

| App | Length TD |
|---|---|
| Skype | 4.9948 |
| Facebook | 6.5112 |
| Game | 3.2843 |
| WhatsApp | 4.2030 |
| Viber | 3.8317 |
| YouTube | 4.8685 |

**5. Defense Evaluation**

We propose in this section a measure to evaluate obfuscation techniques against traffic analysis attacks. This measure describes how good a certain obfuscation model is and compares between different defense systems. We name it Obfuscation Effectiveness (OE) and we define it using two metrics: the attacker confidence, and the data overhead.

The attacker confidence is defined as the estimated ratio of correct app inferences to attempted app inferences by an attacker with no previous knowledge if an obfuscation technique is already deployed. Lesser attacker confidence indicates that the technique is more efficient at defending user privacy. With no protection, attack confidence can reach 1.

Data overhead reflects the padding overhead induced by adding additional dummy bytes by a certain obfuscation technique. It is defined as the ratio of network data sent with and without a certain obfuscation technique. For instance, a bandwidth overhead of 1.5 indicates that applying the technique results in 1.5 times as much bytes traffic sent on unprotected network traffic. This ratio is generally greater than 1. Of course, a smaller data overhead is desirable because additional traffic leads to more network congestion.

A main issue for the robustness of an obfuscation technique is to check how much effort it requires, and how much overhead it induces at the same time. Eventually, we define the obfuscation effectiveness as the product of attacker confidence and the data

overhead. Some obfuscation models have a fixed product but others are adjustable to users' preferences for privacy versus data expenses.

In this context, obfuscation effectiveness is a measure that reflects the model performance as a function of the amount of overhead and secrecy realized. The smaller the obfuscation effectiveness, the more the defense model has good performance in terms of secrecy realized and the resulting data overhead.

We present in Table 16, the obfuscation effectiveness of the Random Forest classifier for some obfuscation algorithms suggested in our previous work, in comparison with OPriv. As expected, in using optimization, OPriv has the smallest value of obfuscation effectiveness among all the other techniques and therefore has the best results.

**Table 16.** Obfuscation effectiveness (OE).

| Obfuscation Model | Version | OE |
|---|---|---|
| AdaptiveMutate [47] | Packet length mutation | 0.369 |
| AdaptiveMutate [47] | IAT mutation | 0.071 |
| AdaptiveMutate [47] | Lengths and IAT mutation | 0.084 |
| Anonymization through prob. distr. [48] | First alg. (to normal distr. of Skype) | 0.087 |
| Anonymization through prob. distr. [48] | First alg. (to Poisson distr. of Viber) | 0.137 |
| Anonymization through prob. distr. [48] | Second algorithm | 0.16 |
| Mutation To similar prob. distr. [49] | - | 0.02 |
| Optimal Analytical Solution [12] | - | 0.019 |
| OPriv | - | 0.015 |

## 6. Discussion and Future Work

The work presented in this study is realistic and precise to implement in real-world practice. It is differential from other past studies by the way that OPriv is coded in an optimization software so that it is immediate and practical. In addition, the mapping of values between original and mutated is realized as length to length and not bin to bin. This implies better precision of our algorithm than those of previous works. The results show the effectiveness of our obfuscation system in balancing performance and security. It ensures privacy protection and reduces bandwidth overhead at the same time. Therefore, OPriv is suitable for all types of network devices and does not need many processing requirements.

Another advantage of our method is that the tables mapping the mutation needed to modify the incoming length into the optimal length can be realized offline. Thus, OPriv can run in a dynamic online manner with minimal latency.

The mapping of the packet lengths from a source app to the corresponding optimal lengths in the target app, Tables 5–10, and the obfuscation results in Table 11 are very compatible with the same mapping and results in our previous work [12]. This evidences the solidarity of the analytical model in [12] without the need for an optimization software. However, it makes it easier to use an available commercial software.

The information metrics presented in this research proved, for the first time, the obfuscation results of our work and of other previous works. This gives significant understanding of the obfuscation results for any thwarting technique. In addition, these metrics can serve as criteria for selecting tunable parameters to achieve the best results of the obfuscation model.

## 7. Conclusions

This paper addresses the privacy protection of network communication against side information based inferential attacks. We study optimal methods to secure network traffic under the scrutiny of an adversary who uses statistical traffic analysis. We present an effective approach, OPriv, to thwart traffic classification optimally: we characterize leaking information, model the attack mathematically, study a heuristic to solve the optimization problem, and evaluate it, using both analytical and experimental models.

Our practical method OPriv is based on solving the nonlinear optimization problem with the IBM ILOG optimization package. The simulation results based on real data traffic confirm that OPriv has reasonable performance in reducing the success rate of statistical traffic analysis attacks. The simulation verifies that this proposed solution performs well on a real data set by reducing the classification accuracy from 91.1% to 1.42% with only 0.17% padding overhead. We also provide in this work novel metrics based on entropy to evaluate, for any obfuscation system, the right choice of features to mutate, as well as the right choice of target applications to mutate to. The mathematical modeling, information theory evaluation, and experimental results presented offer both theoretic and real-world proofs to protect network users' privacy.

**Author Contributions:** Formal analysis, L.C.; Methodology, L.C.; Supervision, A.C. and A.K.; Writing—original draft, L.C.; Writing—review & editing, A.C.and A.K. All authors have read and agreed to the published version of the manuscript.

**Funding:** This research was funded by the AUB University Research Board, and TELUS Corp., Canada.

**Informed Consent Statement:** Not applicable.

**Data Availability Statement:** The data underlying this article will be shared on reasonable request from the corresponding author.

**Conflicts of Interest:** The authors declare no conflict of interest. The funders had no role in the design of the study; in the collection, analyses, or interpretation of data; in the writing of the manuscript, or in the decision to publish the results.

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
