# Peer review of "OPriv: Optimizing Privacy Protection for Network Traffic"

_jsan, doi:10.3390/jsan10030038_

Round 1

Reviewer 1 Report

1. When referring to your previous work, can you briefly introduce the main points and conclusions of your previous work?

2. Is there any other way to solve the problem of opriv's inefficiency on Skype and Facebook.

3. There are some deficiencies in the introduction of padding overhead

Author Response

We would like to thank the Reviewer 1 for his time and valuable comments. We have addressed all of the comments in the detailed response attached, and we have incorporated several modifications into the manuscript.

Reviewer 2 Report

Dear Authors,

Well written paper but there are some minor comments.

  1. Check authors' affiliation and it is same but is has three list. You need to update it. Please talk with editor team and fix it.
  2. In literature review, summary table of Traffic obfuscation techniques looks good. Please summarize the review work either in tabular form  or paragraph by paragraph. References 46-49 are inappropriate self-citations, advised to remove. If needed introduce the concept again.
  3. Table 2 and materials from 122-142 lines are not necessary.
  4. Lack of proper explanation of Figure 2 and 3. It is better to put algorithms rather than codes in figure 2 and 3 for better readability. 
  5. Table 6 to 11 presented without proper explanation. Either remove tables or put proper explanation why these are needed. I am not sure why these tables are there and it is not clear. 
  6. Table 12- lack of proper explanation and discussion.  

Best of luck.

Regards,

Author Response

We would like to thank Reviewer 2 for his time and valuable comments. We have addressed all of the comments in the detailed response attached, and we have incorporated several modifications into the manuscript.

Reviewer 3 Report

In this work, the authors construct a traffic obfuscation against the traffic analysis attack. The authors  make use of CPLEX, an integer programming tool, to decide the obfuscation way. Not only based on the experiment evaluation, the authors propose a mathematical way to formulate their result. Generally speaking, this is an interesting work. My comments are as follows.

  1. Since this is an extension paper from the authors' previous work, the authors should clearly list the extension contributions. Because I cannot get the original work, I am not sure all these five contributions listed in the section 1 are new contributions.
  2. In section 2 table 1, the authors list lots of related works [18-28]. However, in section 2, the authors describe lots of other works, like [29-44]. Why aren't these works listed in table 2?
  3. In section 3.3., what is w? In line 184, w means width, which I do not know what it is, and in line 185, it means the bin size. Are they the same or not?
  4. For me, it is not clear about the concept of "bin". I guess that there are lots of packets for some application and these packets are divided into multiple subgroups, which is called bin where the size is w. However, if I am right, I do not know why a packet should be determined which bin it belongs to since all these bins are for the same application. The authors should give more clear description about their obfuscation approach.
  5. In line 192 item 2, why the authors use 0.1?
  6. Figure 2 and Figure 3 are not clear enough. I suggest that the authors should use texts instead of figures to show their codes.
  7. The padding and chopping operations are not described in this paper. Though they may be trivial, as a reader, I think these should be contained in this paper.
  8. I am wondering if a packet is chopped into multiple packets, should these newly generated packets be re-obfuscated?
  9. In this paper, the authors evaluate their works based on the packet size and the inter-arrival time. However, there are lots of other attributes. I suggest that the authors consider other attributes in their future works.
  10. In section 5, OE is defined as the product of AC and DO without explanation. I suggest the authors to separate OE into AC and DO. Moreover, DO is only considered as the summation of the packet size. Will the chopping overhead be included in DO?

Author Response

We would like to thank Reviewer 3 for his time and valuable comments. We have addressed all of the comments in the detailed response attached, and we have incorporated several modifications into the manuscript.

Round 2

Reviewer 3 Report

All my concerns are well addressed. I have no further questions.